# Study protocol: A cross-sectional survey of clinicians to identify barriers to clinical practice guideline implementation in the assessment and treatment of persistent tic disorders

**Jaclyn M. Martindale**[1]*, **Harini Sarva**[2], **Davide Martino**[3], **Donald L. Gilbert**[4], **Christos Ganos**[5], **Tamara Pringsheim**[6], **Kevin Black**[7], **Irene A. Malaty**[8], all on behalf of the Movement Disorder Society Tic and Tourette Study Group[¶]

1 Department of Neurology, Atrium Health Wake Forest Baptist, Wake Forest University School of Medicine, Winston-Salem, NC, United States of America, 2 Department of Neurology, Division of Neurodegenerative Diseases Weill Cornell Medicine, Parkinson's Disease and Movement Disorders Institute, New York, NY, United States of America, 3 Department of Clinical Neurosciences, University of Calgary and Hotchkiss Brain Institute, Cumming School of Medicine, Calgary, Alberta, Canada, 4 Department of Pediatrics, Division of Neurology, Cincinnati Children's Hospital Medical Center, University of Cincinnati College of Medicine, Cincinnati, OH, United States of America, 5 Department of Neurology, Charité University Medicine Berlin, Berlin, Germany, 6 Department of Clinical Neurosciences, Psychiatry, Pediatrics and Community Health Sciences University of Calgary, Calgary, Alberta, Canada, 7 Departments of Psychiatry, Neurology, Radiology, and Neuroscience, Washington University School of Medicine, St. Louis, MO, United States of America, 8 Department of Neurology, Norman Fixel Institute for Neurological Diseases, University of Florida College of Medicine, Gainesville, FL, United States of America

¶ Membership of the Movement Disorder Society Tic and Tourette Study Group is provided in the Acknowledgments.
* jmartind@wakehealth.edu

## Abstract

### Introduction

Eight members of the International Parkinson's Disease and Movement Disorders Society Tic and Tourette Syndrome Study Group formed a subcommittee to discuss further barriers to practice guideline implementation. Based on expert opinion and literature review, the consensus was that practice variations continue to be quite broad and that many barriers in different clinical settings might negatively influence the adoption of the American Academy of Neurology and the European Society for the Study of Tourette Syndrome published guidelines.

### Objectives

1) To identify how clinical practices diverge from the existing American Academy of Neurology and European Society for the Study of Tourette Syndrome guidelines, and 2) to identify categories of barriers leading to these clinical care gaps.

**Data Availability Statement:** Deidentified research data will be made publicly available when the study is completed and published.

**Funding:** The author(s) received no specific funding for this work.

**Competing interests:** I have read the journal's policy and the authors of this manuscript have the following competing interests: JMM has no conflicts that affect the content of this manuscript. In the past two years, she has received honoraria from the Tourette Association of America. In addition, she participates in research funded by the Tourette Association of America and the American Board of Psychiatry and Neurology. HS has no conflicts that affect the content of this manuscript. Dr. Sarva has done consulting work for Neuroderm, Bluerock, Novo Nordisk, and CALA Health. In addition, she has received clinical trial support from Neuroderm, Bluerock, Prevail, Covance, and NIH. IAM has no conflicts that affect the content of this manuscript. She has participated in research funded by AbbVie, Boston Scientific, Eli Lilly, Neuroderm, and Revance but has no ownership interest in any pharmaceutical company. In addition, she has received travel compensation or honoraria from the Tourette Association of America, Parkinson Foundation, Medscape, Efficient CME, and Cleveland Clinic, and royalties for writing a book with Robert Rose publishers. DM has no conflicts that affect the content of this manuscript. In the past two years, he has received personal compensation for consultancies by Roche, Sunovion, and Merz Pharmaceuticals but has no ownership interest in any pharmaceutical company. He has also received travel compensation or honoraria from the Dystonia Medical Research Foundation of Canada, Movement Disorders Society, and the American Academy of Neurology, and book royalties from Springer-Verlag and Oxford University Press. In addition, he received research support from Ipsen Corporate, Owerko Foundation, Dystonia Medical Research Foundation Canada, Parkinson Canada, and the Michael P Smith Family. DLG has received compensation for expert testimony for the U.S. National Vaccine Injury Compensation Program through the Department of Health and Human Services. He has received payment for medical expert opinions through Advanced Medical/ Teladoc. He has served as a consultant for Applied Therapeutics, Eumentics Therapeutics, and Emalex. He has received research support from the NIH and the DOD. He has received salary compensation through Cincinnati Children's for work as a clinical trial site investigator from Emalex (clinical trial, Tourette Syndrome) and EryDel

## Methods and analysis

This article presents the methodology of a planned cross-sectional survey amongst healthcare professionals routinely involved in the clinical care of patients with persistent tic disorders, aimed at 1) identifying how practices diverge from the published guidelines; and 2) identifying categories of barriers leading to these clinical care gaps. Purposeful sampling methods are used to identify and recruit critical persistent tic disorders stakeholders. The analysis will use descriptive statistics.

## Introduction

Persistent (Chronic) Tic Disorders (PTD), including Tourette Syndrome (TS), are developmental neuropsychiatric disorders characterized by multiple motor and vocal tics present for at least one year beginning before the age of 18 years [1]. It is estimated that PTDs affect up to 3% of children [2]. Approximately 90% of individuals with PTD have at least one co-occurring neuropsychiatric condition, such as attention deficit hyperactivity disorder (ADHD), obsessive-compulsive disorder (OCD), or anxiety [3–7]. The clinical spectrum of PTD is heterogeneous. For most individuals with PTD, tics improve through adolescence; however, persistent moderate-to-severe tics or worsening in adulthood can occur [8–11]. PTD can be associated with poorer quality of life (QOL) and interference in daily living [12–14]. Both tic severity and psychiatric comorbidities can lead to psychosocial, functional, and physical difficulties that can disrupt educational attainment, employment, and social interactions [12, 15–18]. As a result, these patients require not only treatment for their tics but also their comorbidities.

Both the American Academy of Neurology (AAN) [19] and the European Society for the Study of Tourette Syndrome (ESSTS) [20–23] have published guidelines regarding the clinical evaluation and management of PTD. Although these guidelines considered cost, feasibility, and acceptability in their recommendations, there is marked diversity in clinical assessment and management geographically, even amongst movement disorder specialists [24]. For example, these guidelines recommend evaluating tic severity with validated scales; however, only 10% of movement disorder clinicians use standardized scales in clinical practice and vary in which scales. Additionally, although both guidelines recommended behavioral therapy as the primary treatment, accessibility to behavioral therapists globally remains challenging.

These barriers to guideline implementation are reported by healthcare professionals, patients, and families alike. The financial coverage of treatments, accessibility to and cost of specialized PTD care, and additional neuropsychiatric symptoms are the most frequently reported barriers to optimal PTD care [24]. Some studies have specifically assessed patients' and families' perspectives on the existing care model for tics. A systematic review [25] of these studies found several significant barriers to appropriate care identified by people with tics: difficulty in diagnosing patients with TS; fear of stigmatization leading to delays in seeking treatment; concerns about adverse effects of the commonly used medications; concern for potential rebound effect of behavioral therapy leading to worsening of tics or onset of new tics; financial cost of, the time commitment for, and travel distance to behavioral services. In addition, parents of children with tics would have preferred to have more support to help cope with the diagnosis, including active collaboration with school officials, teachers, and pediatricians, which could lead to reduced barriers to receiving optimal care [25].

There is a need to assess the degree to which these clinical guidelines are being followed, contextualize them within diverse practices, and identify barriers encountered so that

(clinical trial, Ataxia Telangiectasia). He has received book/publication royalties from Elsevier, Wolters Kluwer, and the Massachusetts Medical Society. TP has no commercial or financial relationships that could be construed as a potential conflict of interest. TP receives research support from Alberta Health, the Alberta Children's Hospital Research Institute, and the Public Health Agency of Canada. KJB has no conflicts that affect the content of this manuscript. In the past two years, author KJB consulted for SK Life Science, Inc., served as faculty for CME programs by Medscape and Mededicus, and served as an expert rater for the Huntington Study Group, which had contracted with Neurocrine Biosciences. In addition, his institution received research funding from Emalex Biosciences for treatment studies with ecopipam. CG has no conflicts that affect the content of this manuscript. A Freigeist Fellowship of the Volkswagen Stiftung supports him. He has received honoraria from the Movement Disorder Society and BIAL for educational activities. He received honoraria from Biomarine Pharmaceuticals as Ad Hoc Advisory Board. This does not alter our adherence to PLOS ONE policies on sharing data and materials.

strategies can be developed to achieve a standard of care consistently. This article presents the rationale and methodology for a survey of healthcare professionals routinely involved in the clinical care of patients with persistent tic disorders, aimed at identifying existing barriers to the practical implementation of published guidelines into routine clinical care.

## Published guidelines

The AAN and the ESSTS have published guidelines regarding the clinical evaluation and management of PTD based on evidence-based literature, randomized controlled trials (RCTs), and expert opinion. Although these guidelines focus on evidence-based treatments through RCTs, there are scientific and ethical limitations to these studies with the PTD population [26]. For example, limited inclusion or recruitment of individuals with PTD and comorbid neuropsychiatric conditions or concomitant medications in RCTs and the heterogeneity of PTD may limit generalizability to the general PTD population. Additionally, there is limited guidance on combination therapies, optimal duration or the dose of treatment, and treatment indications, all necessary considerations for individualized treatment in routine clinical practice.

The 2019 AAN guidelines provided forty-three total recommendations for optimal care of PTD. A multidisciplinary panel performed a systematic review of RCTs [27]. The panel included nine physicians, two psychologists, and two patient representatives with expertise in child and adult neurology, psychiatry, pediatrics, behavioral treatments, and methodology. For all non-evidence-based factors, including value, availability, risk, and cost, the panel assigned levels of obligation to each recommendation and voted independently and anonymously online using a modified Delphi process. Statistical analysis synthesized the responses, and a consensus was obtained in three rounds of voting. Of the forty-three total recommendations, there were twenty-two level A (must), fifteen level B (should), and nine level C (may) recommendations for the assessment and management of people with PTD [19].

The 2021 ESSTS guidelines consist of four articles summarizing recommendations for assessment [22], psychological interventions [23], pharmacological treatment [20], and deep brain stimulation [28] for PTD. In addition, the ESSTS working group updated and summarized evidence-based literature from the original 2011 guidelines. An online survey of clinical practices and expert opinions completed by 59 ESSTS members in 17 European countries was incorporated into the updated guidelines [21].

Despite these guidelines and the high prevalence of PTD, many patients experience a substantial delay in diagnosis and limited access to effective first-line interventions. For example, a US patient survey by the Tourette Association of America (TAA) found that 78% of adult respondents reported more than a 2-year delay in diagnosis from symptom onset. Several factors include high reliance on specialists to make the diagnosis, difficulty in diagnosing patients with TS, or fear of stigmatization leading to delays in seeking treatment [25, 29].

Both guidelines suggest the importance of proper counseling about the natural history of tics, screening for psychiatric comorbidities, psychosocial education, and the importance of behavioral intervention [23, 30]. Both guidelines recommend using Cognitive Behavioral Intervention for Tics (CBIT) as first-line therapy after several high-quality studies showed a beneficial impact on tic disorders [23, 30]. However, only 36% of children and 25% of adults included in the TAA survey had participated in CBIT. In addition, awareness, accessibility to, the financial cost of, the time commitment for, and travel distance to behavioral services as well as skepticism, were commonly cited barriers to trying CBIT [25, 31].

Regarding medications, the AAN practice guidelines include alpha-agonists, particularly for those with comorbid ADHD; antipsychotic medications were recommended by the AAN when the benefits outweighed the risks [30]. In addition, the ESSTS recommended

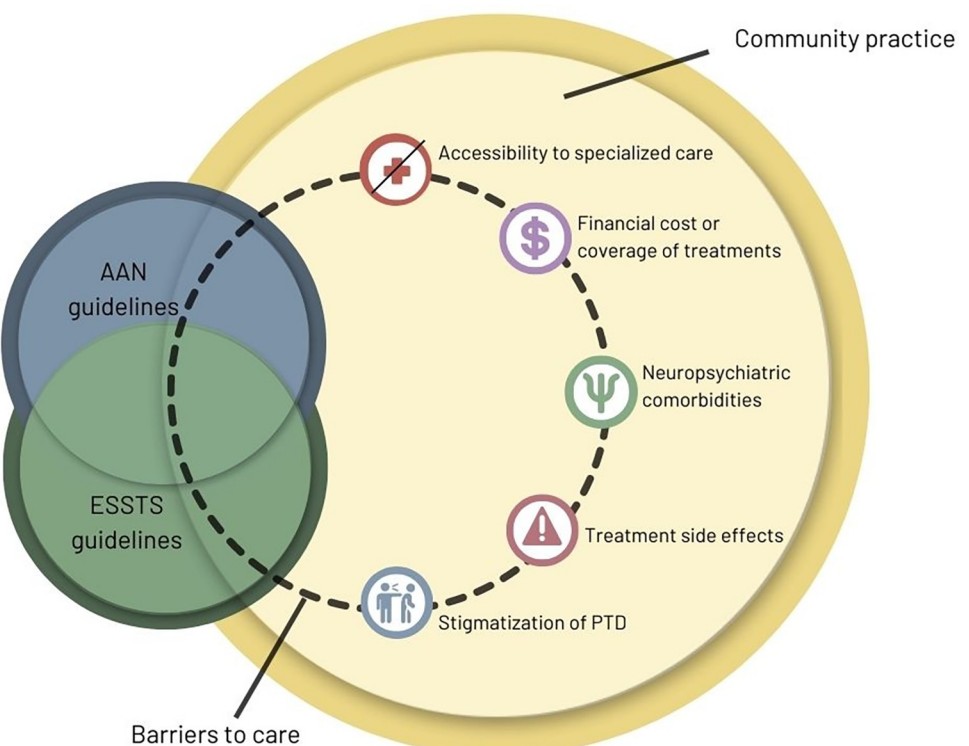

**Fig 1. Conceptualized barriers to guideline implementation in persistent tic disorder care.** AAN: American Academy Neurology; ESSTS: European Society for the Study of Tourette Syndrome; PTD: Persistent Tic Disorder.

aripiprazole as an effective therapy based on two large-scale randomized controlled trials [20]. medication is commonly prescribed, only 44% of children and 47% of adults reported that their symptoms were adequately controlled [13].

Surgical treatments for refractory PTD are an evolving field. For example, with regards to deep brain stimulation (DBS), ESSTS described this therapy as having limited evidence; AAN recommended the use of a multidisciplinary care model for evaluation to confirm the diagnosis, assess psychiatric comorbidities and ensure at least two standard medication trials have failed to produce adequate benefits [28, 30].

Despite the publication of AAN and ESSTS guidelines, the variability of clinical practices and barriers to guideline implementation impede standardized PTD care, Fig 1. Therefore, the authors propose a survey to evaluate how clinical practices diverge from existing guidelines and identify barriers to guideline implementation in PTD. Understanding these barriers provides the initial steps to evaluating and designing interventions to elevate community care of PTD to the existing guidelines.

## Objectives

The objectives of this study are:

1. To identify how clinical practices diverge from the existing AAN and ESSTS guidelines.

2. To identify categories of barriers leading to these clinical care gaps.

## Methods

### Ethics and dissemination

The Wake Forest University Health Sciences (WFUHS) Institutional Review Board IRB00091225 approved the study. Results will be published in a relevant scientific journal and communicated to respondents and relevant institutions.

### Survey development

Members of the International Movement Disorder Society (MDS) Tic and Tourette Syndrome Study Group (TTSSG) initiated a series of discussions regarding whether care defined by the guidelines was being adopted widely. Based on expert opinion and literature review, the consensus was that practice variations continue to be quite broad and that many barriers in different clinical settings might negatively influence the adoption of the AAN and ESSTS guidelines. Following a series of email exchanges between TTSSG members, a subcommittee of eight members was identified, spanning US, Canada, and Germany. This group met by video conference in February 2022 to further discuss barriers to practice guideline implementation. First author JMM led the meeting on February 16, 2022, and a strategic plan was launched for survey development. Patients or the public were not involved in the design of our survey.

A survey was created in Google Documents to allow for easy revisions of multiple collaborators. All authors, highly experienced in subspecialized care for PTD, participated in the survey development. The initial survey covered the collective guidelines from the AAN and ESSTS for both evaluation and management of PTD, practice preferences, frequency of guideline implementation, and perceived barriers to providing optimal care. JMM created the first draft of the survey. Multiple revisions occurred between February through May 2022, altering the survey's context, flow, and length. HS contributed substantially to the development and modifications of the survey and was asked to serve as the second author. The revised survey consisted of seven sections: demographics, clinical assessment, general treatment considerations, psychoeducation, behavioral interventions, pharmacological interventions, and deep brain stimulation. The survey uses branching logic which alters the completion time for each participant, which is around 15–25 minutes. While the survey is lengthier, the context of the survey is critical for understanding current clinical practices and evaluating the barriers to existing guideline implementation. The content of each section is summarized below.

### Demographics

Non-identifying information will be collected as to the specialty of the participant, whether they are a movement disorder specialist, the percentage of PTD patients in their clinical practice, and years of experience in specialty practice. Additionally, practice setting, typical patient age range, and region of practice will be collected.

### Clinical assessment

The survey evaluates what each participant routinely performs for clinical assessment of PTD from collective recommendations of the AAN and ESSTS guidelines. Despite many specialists using guidelines [24], time constraints, access to, and cost of standardized scales may pose challenges in most clinical settings to implementing the comprehensive clinical assessments recommended. For example, most initial consultations for tic disorders are 1–2 hours [32]. Several studies have attempted to define a globalized model of care [21, 24, 32]; however, this section focuses on identifying what should be completed at minimum in the clinical

assessment of persons with PTD. Additionally, we explore clinical assessment preferences and barriers to conducting those assessments.

## General treatment considerations

Given the complexity and heterogeneity of tic disorders, a consensus on indicators for the need to initiate treatments in PTD is difficult to obtain. The 2011 ESSTS guidelines reviewed general considerations of treatment indications in PTD, but no formal recommendations exist on when intervention is necessary [33]. The AAN guidelines state that treatment recommendations pertain only to the situation in which the patient and clinician have determined that treatment is necessary and collaboratively discussed treatment options and priorities. The survey asks what each participant does routinely when considering the treatment of PTD according to the collective clinical guidelines. Additionally, we evaluate what barriers are frequently encountered to following the clinical guidelines, practice preferences of treatment approach, and hierarchical approach to treatment based on the most bothersome symptom.

## Psychoeducation

Both guidelines relay the importance of psychoeducation as part of the management of PTD. Psychoeducation generally refers to the shared knowledge of natural history, etiology, prognosis, and treatment options [23]. The survey evaluates which psychoeducational guideline recommendations are routinely implemented, the barriers to providing such education, and the frequency at which psychoeducation should be reviewed in clinical practice.

## Behavioral interventions

Both guidelines recommend behavioral therapy as the first line for treating TS [19, 21]. The AAN explicitly recommends CBIT above other behavioral therapy forms [19]. The ESSTS included either habit reversal therapy (HRT) or exposure and response prevention (ERP) when psychoeducation alone was ineffective. Access to, cost, and travel distance to trained providers remains a global challenge. Telehealth and internet delivery of behavioral therapy likely increased during the COVID-19 pandemic; however, less is known about the effectiveness of these modalities compared to in-person options, although this is evolving [34, 35]. We ask what the survey participant routinely does when recommending behavioral interventions according to the AAN and ESSTS guidelines. In addition to questioning the barriers to implementing these guidelines, this section also asks for the expert's rank for preference of behavioral interventions.

## Pharmacologic interventions

Existing evidence on direct comparison of pharmacological interventions in TS remains limited. More research is necessary to delineate pharmacological recommendations, particularly in combination with behavioral therapy or comorbid conditions. Moreover, clinical trials must be of sufficient duration to account for the natural fluctuation of tics. Based on a systematic review of randomized controlled trials (RCTS), the AAN concluded there was moderate confidence in the evidence for several medications but did not provide a hierarchical approach of pharmacological agents [19, 27]. The ESSTS recommended aripiprazole for both pediatrics and adults as a first-line agent based on a review of the evidence and expert opinion [20]. This section aims to assess similarities and differences in actual clinical practice, clinical decision-making when choosing a pharmacological agent, and the barriers clinicians face in prescribing treatment regimens for PTD.

### Deep brain stimulation

Deep brain stimulation is evolving to treat severe tics resistant to medical therapies. The optimal surgical target and timing to intervene in a pediatric-onset condition, which may improve over time, remain controversial. There is consensus that a thorough multidisciplinary evaluation is necessary when considering DBS in severe, refractory TS in adults [19, 24, 28, 36]. In addition to the rarity of actual DBS surgery for TS, part of the challenge remains that there needs to be a clear consensus on what is considered treatment-refractory TS [37–39]. Several members of the TTSSG previously conducted a qualitative study to achieve an expert consensus definition of treatment failure [40]. Through the Delphi process, treatment failure was defined on three pillars: lack of efficacy, limited tolerability, and limited adherence. An algorithmic approach to establishing treatment failure was provided for both behavioral and pharmacological interventions. The survey evaluates which of these recommendations are actively pursued by respondents and how they define refractoriness in their clinical practice.

### Open-ended feedback

The survey's final question is an optional open-ended question allowing participants to share any additional barriers or challenges faced in clinical practice when providing care for PTD patients. This can provide additional in-depth insights and allow for identifying emerging themes and subthemes [41].

### Recruitment

Purposeful sampling methods [42, 43] are used to identify and recruit key PTD stakeholders from our research networks, specialty clinics, general neurology and psychiatry clinics, and stakeholder organizations. Clinicians with direct experience treating tic disorders and TS will be asked to participate, focusing on child and adult neurologists and psychiatrists. Recruitment methods include email invitations to providers listed in the Tourette Association of America directory, Movement Disorder Society member update, AAN movement disorder community page, ESSTS newsletter, and social media. The survey will be open from May through July 2023. Data will be collected through REDCap.

### Inclusion criteria

1. Direct experience with treating tic disorders and TS

2. Clinical provider, including a physician or advanced practice provider

### Exclusion criteria

1. Participants who are non-English speaking will be excluded from this study

## Data management and oversight

All centralized study data are stored and managed using an External REDCap database and related tools hosted at Wake Forest University Health Sciences (WFUHS) on a secure server [44]. Data will be maintained in a study database on REDCap. Access to study data will be available to research personnel.

## Analysis

Results will be analyzed initially using descriptive statistics. Then, demographic and clinical variables will be compared between groups using Wilcoxon rank sum tests or chi-square tests, as appropriate. Analyses will use SAS 9.3.

## Discussion

Given the high prevalence of PTD, many patients will not get their care from PTD experts. A significant portion of individuals will receive care from primary doctors, neurologists, and psychiatrists who may have limited PTD expertise. These clinicians face the challenge of limited resources in healthcare communities and patient financial access. This circumstance leads to disparity in care delivered.

The purpose of this survey will be to clarify how current practice differs from the evidence-based AAN and ESSTS guidelines. While the AAN and ESSTS guidelines consider all the practical elements for both physicians and patients in the generation of its recommendations—cost, feasibility, acceptability to patients and expert input, and the real world clear, practical applicability—we do not know empirically the degree to which many of these recommendations are practiced. Practitioner knowledge, available time, and other practical factors likely influence the degree to which these guidelines have been implemented since publication. By discerning how current practice across multiple settings diverges from guideline recommendations, the healthcare community can focus on optimizing education and care to areas where approach diverges from recommendations. This will have widespread implications in focusing the direction of and improving the care provided.

## Limitations

There are some inevitable limitations to this approach. Current evidence to guide treatment in PTD is hampered by inconsistencies in outcomes used, doses of medication employed, and limitations in the duration of studies. As a result, physician responders develop their treatment protocols considering the available data and their experience, which may be influenced by local resource availability and other factors. Still, this allows some framework to guide the best treatment within the current limitations. Furthermore, the questionnaire will be able to deduce additional categories that present barriers to care delivery. Still, more work may be needed to reach a granular level of understanding of the factors that interplay and how these may be overcome. We hope that the effort to identify barriers to guideline implementation will result in opportunities for education, advocacy, and strategy development to improve consistency in care delivered from diverse treatment environments. Furthermore, incorporating a broader range of perspectives, including primary care clinicians, general neurologists, patients, and patient advocates, into developing these strategies would provide valuable insight.

## Supporting information

**S1 File. Barriers to PTD care REDCap survey.**
(PDF)

## Acknowledgments

In addition to the authors, the Movement Disorder Society Tic and Tourette Study Group includes the following members: 1) Mark Hallett, National Institute of Neurological Disorders and Stroke, Bethesda, MD; 2) Andreas Hartmann, Department of Neurology, Hôpital de la

Pitié-Salpêtrière, Paris, France; 3) Mariam Hull, Pediatric Movement Disorders Clinic, Section of Pediatric Neurology and Developmental Neuroscience, Texas Children's Hospital and Baylor College of Medicine, Houston, TX; 4) Andrea Lee, Department of Neurology, University of Kansas Medical Center, Kansas City, KS; 5) Pablo Mir, Unidad de Trastornos del Movimiento Servicio de Neurología y Neurofisiología Clínica, Instituto de Biomedicina de Sevilla Hospital Universitario Virgen del Rocío/Universidad de Sevilla, Seville, Spain; 6) Kirsten Muller-Vahl, Clinic of Psychiatry, Social Psychiatry and Psychotherapy, Hannover Medical School, Carl-Neuberg-Straße, Hannover, Germany; 7) Alexander Muenchau, Institute of Systems Motor Science, University of Lübeck Lübeck Germany; 8) Michael Okun, Department of Neurology, Norman Fixel Institute for Neurological Diseases, Gainesville, FL; 9) Mered Parnes, Pediatric Movement Disorders Clinic, Section of Pediatric Neurology and Developmental Neuroscience, Texas Children's Hospital and Baylor College of Medicine, Houston, TX; 10) Kallol Set; Department of Pediatric Neurology Dayton Children's Hospital, Boonshoft School of Medicine, Wright State University Dayton Ohio; 11) David Shprecher; Banner Sun Health Research Institute, Sun City, AZ; 12) Harvey Singer, Department of Neurology, Johns Hopkins University School of Medicine, Baltimore, MD, 13) Katarzyna Smilowska, Department of Neurology, Regional Specialist Hospital of St. Barbara in Sosnowiec, Poland; 14) Natalia Szejko, Department of Neurology, Medical University of Warsaw, Warsaw, Poland; 15) Daniel van Wamelen, Department of Neuroimaging, King's College London Institute of Psychiatry, Psychology & Neuroscience, London, United Kingdom; 16) Yulia Worbe, ICM, Inserm, CNRS, Department of Neurophysiology, Hôpital Saint Antoine (DMU 6), AP-HP, Sorbonne University, Paris, France.

## Author Contributions

**Conceptualization:** Jaclyn M. Martindale, Harini Sarva, Davide Martino, Donald L. Gilbert, Christos Ganos, Tamara Pringsheim, Kevin Black, Irene A. Malaty.

**Methodology:** Jaclyn M. Martindale, Harini Sarva, Davide Martino, Kevin Black, Irene A. Malaty.

**Writing – original draft:** Jaclyn M. Martindale, Harini Sarva, Donald L. Gilbert, Irene A. Malaty.

**Writing – review & editing:** Jaclyn M. Martindale, Harini Sarva, Davide Martino, Donald L. Gilbert, Christos Ganos, Tamara Pringsheim, Kevin Black, Irene A. Malaty.

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
