## [Decision Letter · Decision Letter 0]

4 May 2023

PONE-D-23-02941

Study protocol: A cross-sectional survey of clinicians to identify barriers to clinical practice guideline implementation in the assessment and treatment of persistent tic disorders

PLOS ONE

Dear Dr. Martindale,

Thank you for submitting your manuscript to PLOS ONE. After careful consideration, we feel that it has merit but does not fully meet PLOS ONE’s publication criteria as it currently stands. Therefore, we invite you to submit a revised version of the manuscript that addresses the points raised during the review process.

Your Study Protocol has been assessed by four reviewers, and their comments are available below. They appreciate the importance of your research question, but also provided suggestions to improve the reporting and presentation of your study. Please carefully address all concerns raised.

We look forward to receiving your revised manuscript.

Kind regards,

Dario Ummarino, PhD

Senior Editor

PLOS ONE

Journal Requirements:

“I have read the journal's policy and the authors of this manuscript have the following competing interests:

JMM has no conflicts that affect the content of this manuscript. In the past two years, she has received honoraria from the Tourette Association of America. In addition, she participates in research funded by the Tourette Association of America and the Child Neurology Foundation.

HS has no conflicts that affect the content of this manuscript.  Dr. Sarva has done consulting work for Neuroderm, Bluerock, Novo Nordisk, and CALA Health. In addition, she has received clinical trial support from Neuroderm, Bluerock, Prevail, Covance, and NIH. 

IAM has no conflicts that affect the content of this manuscript.  She has participated in research funded by AbbVie, Boston Scientific, Eli Lilly, Neuroderm, and Revance but has no ownership interest in any pharmaceutical company.  In addition, she has received travel compensation or honoraria from the Tourette Association of America, Parkinson Foundation, Medscape, Efficient CME, and Cleveland Clinic, and royalties for writing a book with Robert Rose publishers. 

DM has no conflicts that affect the content of this manuscript. In the past two years, he has received personal compensation for consultancies by Roche, Sunovion, and Merz Pharmaceuticals but has no ownership interest in any pharmaceutical company. He has also received travel compensation or honoraria from the Dystonia Medical Research Foundation of Canada, Movement Disorders Society, and the American Academy of Neurology, and book royalties from Springer-Verlag and Oxford University Press. In addition, he received research support from Ipsen Corporate, Owerko Foundation, Dystonia Medical Research Foundation Canada, Parkinson Canada, and the Michael P Smith Family.

DLG has received compensation for expert testimony for the U.S. National Vaccine Injury Compensation Program through the Department of Health and Human Services. He has received payment for medical expert opinions through Advanced Medical/Teladoc. He has served as a consultant for Applied Therapeutics, Eumentics Therapeutics, and Emalex. He has received research support from the NIH and the DOD. He has received salary compensation through Cincinnati Children’s for work as a clinical trial site investigator from Emalex (clinical trial, Tourette Syndrome) and EryDel (clinical trial, Ataxia Telangiectasia). He has received book/publication royalties from Elsevier, Wolters Kluwer, and the Massachusetts Medical Society.

TP has no commercial or financial relationships that could be construed as a potential conflict of interest. TP receives research support from Alberta Health, the Alberta Children’s Hospital Research Institute, and the Public Health Agency of Canada.

KJB has no conflicts that affect the content of this manuscript. In the past two years, author KJB consulted for SK Life Science, Inc., served as faculty for CME programs by Medscape and Mededicus, and served as an expert rater for the Huntington Study Group, which had contracted with Neurocrine Biosciences. In addition, his institution received research funding from Emalex Biosciences for treatment studies with ecopipam.

CG has no conflicts that affect the content of this manuscript. A Freigeist Fellowship of the Volkswagen Stiftung supports him. He has received honoraria from the Movement Disorder Society and BIAL for educational activities. He received honoraria from Biomarine Pharmaceuticals as Ad Hoc Advisory Board.“

3. One of the noted authors is a group or consortium “Movement Disorder Society Tic and Tourette Study Group”. In addition to naming the author group, please list the individual authors and affiliations within this group in the acknowledgments section of your manuscript. Please also indicate clearly a lead author for this group along with a contact email address.

Reviewers' comments:

Reviewer's Responses to Questions

**Comments to the Author**

1. Does the manuscript provide a valid rationale for the proposed study, with clearly identified and justified research questions?

Reviewer #1: Yes

Reviewer #2: Yes

Reviewer #3: Yes

Reviewer #4: Yes

2. Is the protocol technically sound and planned in a manner that will lead to a meaningful outcome and allow testing the stated hypotheses?

Reviewer #1: Yes

Reviewer #2: Partly

Reviewer #3: Yes

Reviewer #4: Yes

3. Is the methodology feasible and described in sufficient detail to allow the work to be replicable?

Reviewer #1: Yes

Reviewer #2: Yes

Reviewer #3: Yes

Reviewer #4: Yes

4. Have the authors described where all data underlying the findings will be made available when the study is complete?

Reviewer #1: No

Reviewer #2: Yes

Reviewer #3: Yes

Reviewer #4: Yes

5. Is the manuscript presented in an intelligible fashion and written in standard English?

Reviewer #1: Yes

Reviewer #2: Yes

Reviewer #3: Yes

Reviewer #4: Yes

6. Review Comments to the Author

You may also provide optional suggestions and comments to authors that they might find helpful in planning their study.

Reviewer #1: This is a well written protocol that is likely to be followed by many researchers to conduct future research in this topic.

Apart from a few repetitive words in certain sections, the protocol is well written and easy to follow.

it might be useful to discus the modified Delhi process briefly in line 105

It might also be useful to discuss who was involved in the panel mentioned in line 106.

Reviewer #2: It is a good study, and certainly generates relevant questions for practicing providers who deal with TIC disorder. I'm concerned about the length of the survery which is 30 pages long, and wonder if the providers would have enough time from their clinical responsibilities to be able to adequately complete the survey. May be creating a timeline for collecting the surveys might help with the timely analysis of the data.

Reviewer #3: The study protocol design aims to identify how clinical practices differ from existing AAN and ESSTS guidelines and the categories of barriers leading to these clinical care gaps. The study protocol includes seven sections, covering demographics, clinical assessment, general treatment considerations, psychoeducation, behavioral interventions, pharmacological interventions, and deep brain stimulation. The survey will be published in a relevant scientific journal and communicated to respondents and relevant institutions. The study protocol design is well-written and provides sufficient details for conducting the research.

Here are some suggestions

It is unclear how the survey will be distributed to potential participants. Will it be sent to all movement disorder specialists, or a specific subset? How will participants be recruited?

The survey development process seems to rely heavily on the input of a small group of experts. It may be beneficial to incorporate the perspectives of a broader range of clinicians and patients, as they may have different insights on barriers to guideline implementation.

It is unclear whether the survey will include any open-ended questions or opportunities for participants to provide qualitative feedback. Including these types of questions may provide more in-depth insights into the barriers and challenges faced by clinicians.

Reviewer #4: Congratulations for authors for this scholarly article. The manuscript draws up a comprehensive database search to find research papers. Moreover, limitations are clearly discussed.

7. PLOS authors have the option to publish the peer review history of their article (what does this mean?). If published, this will include your full peer review and any attached files.

Reviewer #1: **Yes: **Lakshit Jain MD

Reviewer #2: No

Reviewer #3: No

Reviewer #4: **Yes: **Pratik Bahekar

---

## [Author Response · Author response to Decision Letter 0]

6 Jun 2023

Dear Reviewers, 

Thank you for your suggestions for revisions. We have incorporated all of the revisions into the manuscript. 

 - The style has been adjusted according the templates provided. 

 - Figure nomenclature has been adjusted

2. Thank you for stating the following in the Competing Interests section

 - Confirmed it does not altered adherence to PLOS One policies 

 - Added recommended statement on cover letter

 - Updated competing interests’ statement on cover letter

3. One of the noted authors is a group or consortium “Movement Disorder Society Tic and Tourette Study Group”. In addition to naming the author group, please list the individual authors and affiliations within this group in the acknowledgments section of your manuscript. Please also indicate clearly a lead author for this group along with a contact email address.

 - All of the authors are submitting on behalf of the study group. The first and corresponding author is the lead author for this group. Added the word all in author list to provide clarification. 

 - Added affiliations to author list in acknowledgments

 - Moved to methods 154-157

 - Also removed from abstract

 - Reviewed references

6. It might be useful to discuss the modified Delhi process briefly in line 105

 - Expanded upon this in lines 105-108

7. It might also be useful to discuss who was involved in the panel mentioned in line 106.

 - Expanded upon this in lines 103-105

8. I'm concerned about the length of the survey which is 30 pages long, and wonder if the providers would have enough time from their clinical responsibilities to be able to adequately complete the survey. 

 - Discussed in lines 178-181. 

9. May be creating a timeline for collecting the surveys might help with the timely analysis of the data. 

 - Added in line 256

10. It is unclear how the survey will be distributed to potential participants. Will it be sent to all movement disorder specialists, or a specific subset? How will participants be recruited?

 - Expanded upon in lines 252-256

11. The survey development process seems to rely heavily on the input of a small group of experts. It may be beneficial to incorporate the perspectives of a broader range of clinicians and patients, as they may have different insights on barriers to guideline implementation. 

 - Excellent point, we did discuss this. Added in lines 300-302

12. It is unclear whether the survey will include any open-ended questions or opportunities for participants to provide qualitative feedback. Including these types of questions may provide more in-depth insights into the barriers and challenges faced by clinicians

 - Added an open-ended optional question at the end of the survey as an opportunity for participants to provide feedback. 

Thank you for your review of this manuscript. 

Sincerely, 

Jaclyn M. Martindale

---

## [Decision Letter · Decision Letter 1]

26 Jun 2023

Study protocol: A cross-sectional survey of clinicians to identify barriers to clinical practice guideline implementation in the assessment and treatment of persistent tic disorders

PONE-D-23-02941R1

Dear Dr. Martindale,

We’re pleased to inform you that your manuscript has been judged scientifically suitable for publication and will be formally accepted for publication once it meets all outstanding technical requirements.

Kind regards,

Tarik A. Rashid, PhD

Academic Editor

PLOS ONE

Additional Editor Comments (optional):

Reviewers' comments:

Reviewer's Responses to Questions

**Comments to the Author**

1. Does the manuscript provide a valid rationale for the proposed study, with clearly identified and justified research questions?

Reviewer #1: Yes

2. Is the protocol technically sound and planned in a manner that will lead to a meaningful outcome and allow testing the stated hypotheses?

Reviewer #1: Yes

3. Is the methodology feasible and described in sufficient detail to allow the work to be replicable?

Reviewer #1: Yes

4. Have the authors described where all data underlying the findings will be made available when the study is complete?

Reviewer #1: Yes

5. Is the manuscript presented in an intelligible fashion and written in standard English?

Reviewer #1: Yes

6. Review Comments to the Author

You may also provide optional suggestions and comments to authors that they might find helpful in planning their study.

Reviewer #1: The authors have made the appropriate changes as requested, and all the concerns raised have been responded to appropriately. I have no further concerns in regards to this protocol.

7. PLOS authors have the option to publish the peer review history of their article (what does this mean?). If published, this will include your full peer review and any attached files.

Reviewer #1: **Yes: **Lakshit Jain MD

---

## [Editor Report · Acceptance letter]

10 Jul 2023

PONE-D-23-02941R1 

Study protocol: A cross-sectional survey of clinicians to identify barriers to clinical practice guideline implementation in the assessment and treatment of persistent tic disorders 

Dear Dr. Martindale:

I'm pleased to inform you that your manuscript has been deemed suitable for publication in PLOS ONE. Congratulations! Your manuscript is now with our production department. 

Kind regards, 

on behalf of

Dr. Tarik A. Rashid 

Academic Editor

PLOS ONE